# Trust Region Policy Optimization with Optimal Transport Discrepancies: Duality and Algorithm for Continuous Actions

**Antonio Terpin**\*
Automatic Control Laboratory
ETH Zürich
aterpin@ethz.ch

**Nicolas Lanzetti**\*
Automatic Control Laboratory
ETH Zürich
lnicolas@ethz.ch

**Batuhan Yardim**
Dept. of Computer Science
ETH Zürich
alibatuhan.yardim@ethz.ch

**Florian Dörfler**
Automatic Control Laboratory
ETH Zürich
dorfler@ethz.ch

**Giorgia Ramponi**
Dept. of Computer Science
ETH AI Center
giorgia.ramponi@ai.ethz.ch

## Abstract

Policy Optimization (PO) algorithms have been proven particularly suited to handle the high-dimensionality of real-world continuous control tasks. In this context, Trust Region Policy Optimization methods represent a popular approach to stabilize the policy updates. These usually rely on the Kullback-Leibler (KL) divergence to limit the change in the policy. The Wasserstein distance represents a natural alternative, in place of the KL divergence, to define trust regions or to regularize the objective function. However, state-of-the-art works either resort to its approximations or do not provide an algorithm for continuous state-action spaces, reducing the applicability of the method. In this paper, we explore optimal transport discrepancies (which include the Wasserstein distance) to define trust regions, and we propose a novel algorithm – Optimal Transport Trust Region Policy Optimization (OT-TRPO) – for continuous state-action spaces. We circumvent the infinite-dimensional optimization problem for PO by providing a one-dimensional dual reformulation for which strong duality holds. We then analytically derive the optimal policy update given the solution of the dual problem. This way, we bypass the computation of optimal transport costs and of optimal transport maps, which we implicitly characterize by solving the dual formulation. Finally, we provide an experimental evaluation of our approach across various control tasks. Our results show that optimal transport discrepancies can offer an advantage over state-of-the-art approaches.

## 1 Introduction

Reinforcement Learning (RL) has achieved outstanding results in numerous fields, from resource management [16], recommendation systems [42], and optimization of chemical reactions [44], to video-games [18, 43, 10] and board games [39], without sparing the world's champion of GO [32]. Many of these successful applications rely on Policy Optimization (PO) algorithms, a family of RL methods that are particularly suited to handle the high-dimensionality of real-world control tasks. PO algorithms approach the RL setting as an optimization problem in the policy space. In this context, the main challenge is to provide policy improvement guarantees. One remarkable option in this direction

---

\*Equal contribution.

36th Conference on Neural Information Processing Systems (NeurIPS 2022).

is represented by Trust Region Policy Optimization (TRPO) [29], which constrains the optimization problem to policies that are "close" to the current one, whereby the Kullback-Leibler (KL) divergence is used as a similarity measure. Nevertheless, "closeness" in the policy space can also be quantified via other functions. Recent work [22, 33, 20] proposed to replace the KL divergence with the Wasserstein distance, a particular instance of optimal transport discrepancy (or cost). Besides being very natural and expressive, optimal transport discrepancies enjoy powerful topological, differential, geometrical, computational, and statistical features and guarantees [37, 3, 14]. In particular, (i) optimal transport discrepancies allow us to compare probability measures (and thus policies) not sharing the same support (for which the KL divergence is infinity); and (ii) they encapsulate the geometry encoded by the transport cost in the action space: the discrepancy between two actions coincides with the discrepancy between the corresponding deterministic policies (whereas the KL divergence is again infinity). These reasons make optimal transport discrepancies particularly attractive for RL. However, the mere evaluation of optimal transport discrepancies entails solving a transportation problem (e.g., see [23, 34]), which poses significant computational challenges for its deployment. Most of the previous work on the topic [22, 20] overcomes the computational burden via approximation, effectively changing the original problem. Conversely, [33] proposes two algorithms to solve the PO problem exactly, studying the trust regions described via the Wasserstein distance and the Sinkhorn divergence. However, their analysis is limited to discrete (and finite) settings. In our work, we consider optimal transport discrepancies to construct the trust region in settings where actions and states take value in general compact Polish spaces. This allows tackling many applications of interest involving continuous domains such as physical control tasks. We derive and leverage a dual reformulation of the PO problem to ensure an optimal policy update within the trust region, without any additional need for line searches (conversely to [29]). We circumvent the computation of optimal transport discrepancies via an analytical expression of the transport maps, which are characterized thanks to the dual reformulation. Notably, our analysis enables a practical and efficient algorithm that encompasses both discrete and continuous settings.

**Contributions.** Our contributions are summarized as follows:

1. We derive the dual of the optimal transport trust region policy optimization problem and we show that strong duality holds for general compact metric state-action spaces. We further characterize the optimal policy update given the solution to the dual problem. We show that policy updates can result in monotonic improvement of the performance function.
2. We propose a novel PO algorithm for continuous spaces, **O**ptimal **T**ransport **T**rust **R**egion **P**olicy **O**ptimization (OT-TRPO). Herein, we leverage the derived duality theory to provide policy updates that satisfy the optimal transport discrepancy constraint while circumventing its computation.
3. We conduct experiments in several RL benchmarks in both discrete and continuous state-action spaces, comparing our method to state-of-the-art approaches. Our results show the effectiveness of our approach for PO and the benefits of using optimal transport discrepancies.

## 2 Related Works

Optimal transport, and in particular the Wasserstein distance, has found various applications in RL and in particular in PO algorithms. In this section, we discuss the most relevant for our work; a broader overview is postponed to Appendix A.1. In [22], the authors propose Behavior Guided Policy Gradient (BGPG), whereby they replace the KL divergence trust region from TRPO [29] by a Wasserstein distance penalty in a behavioral space. Although our alternating procedure in Section 5 may resemble the approach of [22] in spirit, our approach is fundamentally different; cfr. [22, Algorithms 1 and 3] with Algorithm 1. In [20], the authors further suggested incorporating additional information about the local behavior of policies encapsulated in the so-called Wasserstein Information Matrix, in the attempt to speed up the PO using a Wasserstein Natural Policy Gradient (WNPG). However, these approaches are relatively slow, compared to the traditional Proximal Policy Optimization (PPO) and TRPO. Conversely to our work, they do not build on the idea of trust regions: we instead guarantee that the policy update is "close" to the previous one, where the "closeness" is defined via an optimal transport discrepancy (e.g., the Wasserstein distance). Accordingly, the closest related work to ours is the recent paper [33] which studied Wasserstein Policy Optimization (WPO) for discrete action spaces. In contrast to [33], the present work addresses the general setting of compact Polish spaces encompassing the cases of continuous and discrete state-action spaces as particular cases. While we also adopt a duality approach, our level of generality induces many challenges compared to the discrete action space setting (see Remark 2), and it is compatible even with non-direct policy

parametrizations. Finally, our work is closely connected with Wasserstein Distributionally Robust Optimization (DRO) [8, 19, 4, 25, 13]. Albeit our duality results are inspired from this literature, DRO is concerned with quantifying the worst-case risk of a cost functional over an ambiguity set of probability measures, which is a fundamentally different setting; see Remark 1. Conversely to all the previous work, we show how to perform exact optimal transport-based TRPO in continuous settings: we exploit optimal transport theory to circumvent the computational burden of evaluating optimal transport discrepancies while still performing exact policy updates within the trust regions. Accordingly, to the best of our knowledge, our practical algorithm is completely novel.

## 3  Preliminaries

We briefly introduce useful background and notation for the remainder of the paper.

**Notation.** For every Polish space $\mathcal{X}$ (i.e., completely metrizable separable topological space), the set of Borel probability measures on $\mathcal{X}$ is denoted by $\mathcal{P}(\mathcal{X})$. The Dirac measure at some point $x \in \mathcal{X}$ is denoted by $\delta_x$. Given two Polish spaces $\mathcal{X}, \mathcal{Y}$, a Borel probability measures $\mu \in \mathcal{P}(\mathcal{X})$, and a Borel map $T : \mathcal{X} \to \mathcal{Y}$, the pushforward measure of $\mu$, denoted by $T_{\#}\mu$, is defined by $(T_{\#}\mu)(A) := \mu(T^{-1}(A))$ for all $A \in \mathcal{B}(\mathcal{Y})$, where $\mathcal{B}(\mathcal{Y})$ is the collection of Borel subsets of $\mathcal{Y}$. The set of probability measures on a finite set $\mathcal{X}$ coincides with the probability simplex and will also be denoted by $\mathcal{P}(\mathcal{X})$. For a given function $f : \mathcal{X} \to \mathbb{R}$, the notation $\|f\|_{\infty}$ refers to $\sup_{x \in \mathcal{X}} |f(x)|$.

**Markov Decision Process.** We consider an infinite-horizon discounted Markov Decision Process (MDP) [24] $\mathcal{M} = (\mathcal{S}, \mathcal{A}, \mathrm{P}, r, \rho, \gamma)$, where $\mathcal{S}$ is the state space, $\mathcal{A}$ is the action space, $\mathrm{P} : \mathcal{S} \times \mathcal{A} \times \mathcal{S} \to \mathbb{R}_{\geq 0}$ is the state transition probability kernel, $r : \mathcal{S} \times \mathcal{A} \to \mathbb{R}$ is the reward function, $\rho$ is the initial state probability distribution, and $\gamma \in [0, 1)$ is the discount factor. A randomized stationary Markovian policy, which we will simply call a policy in the rest of the paper, is a mapping $\pi : \mathcal{S} \to \mathcal{P}(\mathcal{A})$ specifying for each $s \in \mathcal{S}$ a probability measure over the set of actions $\mathcal{A}$ by $\pi(\cdot|s) \in \mathcal{P}(\mathcal{A})$. The set of all policies is denoted by $\Pi$. Each policy $\pi \in \Pi$ induces a discrete-time Markov reward process $\{(s_t, r(s_t, a_t))\}_{t \in \mathbb{N}}$, where $s_t \in \mathcal{S}$ represents the state of the system at time $t$ and $r(s_t, a_t)$ corresponds to the reward received when executing action $a_t \in \mathcal{A}$ in state $s_t$. We denote by $\mathbb{P}_{\rho,\pi}$ the probability distribution of the Markov chain $(s_t, a_t)$ issued from the MDP controlled by the policy $\pi$ with initial state distribution $\rho$. The associated expectation is denoted by $\mathbb{E}_{\rho,\pi}$ and the notation $\mathbb{E}_{\pi}$ is used whenever there is no dependence on $\rho$. The state-value function $V^{\pi} : \mathcal{S} \to \mathbb{R}$ and the action-value function $Q^{\pi} : \mathcal{S} \times \mathcal{A} \to \mathbb{R}$ are defined for all $s \in \mathcal{S}, a \in \mathcal{A}$ by $V^{\pi}(s) := \mathbb{E}_{\pi}[\sum_{t=0}^{\infty} \gamma^t r(s_t, a_t)|s_0 = s]$ and $Q^{\pi}(s,a) := \mathbb{E}_{\pi}[\sum_{t=0}^{\infty} \gamma^t r(s_t, a_t)|s_0 = s, a_0 = a]$. We also define the advantage function $A^{\pi} : \mathcal{S} \times \mathcal{A} \to \mathbb{R}$ by $A^{\pi}(s,a) := Q^{\pi}(s,a) - V^{\pi}(s)$. Given an initial state probability distribution $\rho$, our goal is to find a policy $\pi$ maximizing the expected long-term return

$$J(\pi) := \mathbb{E}_{\rho,\pi}\left[\sum_{t=0}^{\infty} \gamma^t r(s_t, a_t)\right],$$

which is well-defined when, e.g., the reward function is bounded. To solve this PO problem, we only have access to the observed state, action, and reward $s_t, a_t, r_t$ at each time step $t$, whereas the state transition kernel $\mathrm{P}$ is unknown. When the state and action spaces ($\mathcal{S}$ and $\mathcal{A}$) are finite, an optimal policy $\pi^*$ is guaranteed to exist. When $\mathcal{S}$ and $\mathcal{A}$ are continuous, a (measurable) optimal policy is also guaranteed to exist (see [24, Theorem 6.11.11, p. 262]) under appropriate assumptions on the state and action spaces, the reward function and the transition kernel; we will explicit these later on. In this paper, we focus on the continuous state-action space setting and comment on the discrete (non necessarily finite) setting as a special case.

**Optimal transport.** Consider a Polish space $\mathcal{X}$ and a continuous non-negative function $c : \mathcal{X} \times \mathcal{X} \to \mathbb{R}_{\geq 0}$, referred to as *transport cost*. Let $\mu, \nu \in \mathcal{P}(\mathcal{X})$ and define the set of joint probability measures on $\mathcal{X} \times \mathcal{X}$ with marginals $\mu$ and $\nu$:

$$\Gamma(\mu, \nu) := \{\gamma \in \mathcal{P}(\mathcal{X} \times \mathcal{X}) : \gamma(A \times \mathcal{X}) = \mu(A), \gamma(\mathcal{X} \times B) = \nu(B) \ \forall A, B \in \mathcal{B}(\mathcal{X})\}.$$

We define the *optimal transport discrepancy* on $\mathcal{P}(\mathcal{X})$ for every probability measures $\mu$ and $\nu$ by

$$C(\mu, \nu) := \min_{\gamma \in \Gamma(\mu, \nu)} \int_{\mathcal{X} \times \mathcal{X}} c(x, x') \, \mathrm{d}\gamma(x, x'). \tag{1}$$

Notice that this definition is valid for both discrete and continuous measures. When $c = d^p$, where $d$ is a distance on $\mathcal{X}$ and $p \geq 1$, then $C(\mu, \nu)^{1/p}$ reduces to the celebrated type-$p$ Wasserstein

distance [37, 2]. In our PO context, we will use this discrepancy to compare two probability measures $\pi(\cdot|s) \in \mathcal{P}(\mathcal{A})$ and $\tilde{\pi}(\cdot|s) \in \mathcal{P}(\mathcal{A})$ for every $s \in \mathcal{S}$, where $\pi, \tilde{\pi} \in \Pi$ are two policies.

# 4   Optimal Transport for Trust Region Policy Optimization

In this section, we study the TRPO algorithm with a trust region defined using an optimal transport discrepancy as a measure of closeness between policies. We prove that the arising optimization problem admits an amenable dual reformulation. Importantly, we show that, given the dual optimal solution, the primal solution has an analytical expression, which can lead to monotonic improvements of the performance index.

## 4.1   Policy iteration algorithm with optimal transport-based trust regions

By the policy difference lemma [11, Lemma 6.1], the difference between the expected returns of two policies $\pi, \tilde{\pi} \in \Pi$ reads

$$J(\tilde{\pi}) = J(\pi) + \int_{\mathcal{S}} \int_{\mathcal{A}} A^\pi(s, a) \mathrm{d}\tilde{\pi}(a|s) \mathrm{d}\rho_{\tilde{\pi}}(s), \tag{2}$$

where $\rho_{\tilde{\pi}}$ is the discounted state-occupancy measure [11]. The complex dependency of the discounted visitation frequency $\rho_{\tilde{\pi}}$ on the policy $\tilde{\pi}$ hampers the direct optimization of (2); see [29, Sec. 2]. Following previous work, we consider instead a local approximation of the expected return $J$, defined by

$$L_\pi(\tilde{\pi}) := J(\pi) + \int_{\mathcal{S}} \int_{\mathcal{A}} A^\pi(s, a) \mathrm{d}\tilde{\pi}(a|s) \mathrm{d}\rho_\pi(s). \tag{3}$$

Observe that this approximation uses the discounted state-occupancy measure $\rho_\pi$ (which can be estimated) instead of $\rho_{\tilde{\pi}}$ (see (2)). In other words, the influence of a policy change on the discounted state-occupancy measure is neglected. Moreover, this surrogate function coincides with the expected return $J$ when $\tilde{\pi} = \pi$. Then, (3) motivates a policy update rule maximizing at each time step the approximation $L_\pi(\tilde{\pi})$ over $\tilde{\pi}$, where $\pi$ is the current policy that we want to improve upon (see also [33, Section 2, Eq. (1)]). To ensure stability of the update, we conservatively update the policy using a discrepancy constraint between the current and the new one. Unlike TRPO, we do not use the KL divergence to define the trust region, but instead an optimal transport discrepancy. Then, at each time step, our method solves

$$\begin{aligned} &\sup_{\tilde{\pi} \in \Pi} \int_{\mathcal{S}} \int_{\mathcal{A}} A^\pi(s, a) \mathrm{d}\tilde{\pi}(a|s) \mathrm{d}\rho_\pi(s), \\ &\text{s.t. } \tilde{\pi} \in \mathcal{T}_\varepsilon(\pi) := \left\{ \tilde{\pi} \in \Pi : \int_{\mathcal{S}} C(\pi(\cdot|s), \tilde{\pi}(\cdot|s)) \mathrm{d}\rho_\pi(s) \leq \varepsilon \right\}, \end{aligned} \tag{P}$$

where $\varepsilon > 0$ is a parameter defining the radius of the trust region $\mathcal{T}_\varepsilon(\pi)$. Similarly to [29, Eq. (12)] and [33, Problem (4)], we consider the *average* optimal transport discrepancy over the state space as optimization constraint. Accordingly, the OT-TRPO policy optimization results from iteratively solving Problem (P).

## 4.2   Dual of the trust-region constrained problem (P)

Problem (P) is intractable for two main reasons. First, as soon as the state or action space is continuous, it is an infinite-dimensional optimization problem. Second, the mere evaluation of the trust-region constraint (e.g., for line search as in TRPO [29]) needs (possibly) infinitely many computations of the optimal transport discrepancy, which is itself already challenging to estimate. However, inspired by prior works on Wasserstein DRO [8, 19, 41, 4], we show that problem (P) admits a tractable one-dimensional convex dual reformulation. This duality theorem is the cornerstone of the design of our algorithm. Before stating the result, we make the following assumptions.

**Assumption 1.** The state space $\mathcal{S}$ is a compact subset of an Euclidean space, the action space $\mathcal{A}$ is a compact subset of a Polish space, the reward function $r$ is a continuous function and for every continuous function $w$ on $\mathcal{S}$, $\int_{\mathcal{S}} w(u) \mathrm{dP}(u|s, a)$ is continuous in both $s$ and $a$.

Under this assumption, there exists an optimal measurable (stationary) policy to the PO problem formulated in Section 3. We refer the reader to [24, Theorem 6.11.11, p. 262] for a statement of this result and milder assumptions. In particular, our duality result continues to hold if $\mathcal{S}$ is a compact Polish space (i.e., not necessarily Euclidean).

**Assumption 2.** For every policy $\pi \in \Pi$, the advantage function $A^\pi : \mathcal{S} \times \mathcal{A} \to \mathbb{R}$ is continuous. Moreover, the transport cost $c : \mathcal{A} \times \mathcal{A} \to \mathbb{R}_{\geq 0}$ is continuous and satisfies $c(a,a) = 0$ for all $a \in \mathcal{A}$.

In the next theorem, we show that under these assumptions Problem (P) admits a dual reformulation for which strong duality holds.

**Theorem 1** (Dual formulation). *For every $\varepsilon > 0$ and for every policy $\pi \in \Pi$, under Assumptions 1 and 2 the following strong duality result holds:*

$$\max_{\tilde{\pi} \in \Pi} \left\{ \int_{\mathcal{S}} \int_{\mathcal{A}} A^\pi(s,a) \, \mathrm{d}\tilde{\pi}(a|s) \mathrm{d}\rho_\pi(s) : \int_{\mathcal{S}} C(\pi(\cdot|s), \tilde{\pi}(\cdot|s)) \mathrm{d}\rho_\pi(s) \leq \varepsilon \right\} \tag{P}$$

$$= \min_{\lambda \geq 0} \left\{ \lambda \varepsilon + \int_{\mathcal{S}} \int_{\mathcal{A}} \max_{a' \in \mathcal{A}} \{ A^\pi(s,a') - \lambda c(a,a') \} \mathrm{d}\pi(a|s) \mathrm{d}\rho_\pi(s) \right\}. \tag{D}$$

*Moreover, the primal and dual problems* (P) *and* (D) *admit a maximizer and a minimizer, respectively.*

Remarkably, Problem (D) is one-dimensional and convex, and it only involves the current policy $\pi$, advantage function $A^\pi$, and visitation frequency $\rho_\pi$. The proof of Theorem 1 is constructive. In particular, we derive a closed-form solution of problem (P) as a function of the optimal Lagrange multiplier $\lambda^*$ solving problem (D) and the policy $\pi$ defining the problem. Even if the closed-form policy is part of Theorem 1 and its proof, we present it separately for clarity and for later reference. To do so, we introduce some additional notation, which is instrumental to derive a practical algorithm (similarly to [33]).

Under Assumptions 1 and 2, define for every $\lambda \geq 0$ the $\lambda$-regularized advantage $\Phi_\lambda : \mathcal{S} \times \mathcal{A} \to \mathbb{R}$ and its associated set of maximizers for every $s \in \mathcal{S}, a \in \mathcal{A}$ as follows:

$$\begin{aligned} \Phi_\lambda(s,a) &:= \max_{a' \in \mathcal{A}} \{ A^\pi(s,a') - \lambda c(a,a') \}, \\ \mathcal{D}_\lambda(s,a) &:= \arg\max_{a' \in \mathcal{A}} \{ A^\pi(s,a') - \lambda c(a,a') \}. \end{aligned} \tag{4}$$

**Corollary 2** (Optimal policy). *Under the setting and assumptions of Theorem 1, for any policy $\pi \in \Pi$, let $\lambda^* \geq 0$ be the minimizer of the dual problem* (D). *Then, the following statements hold:*

*1. For every $\lambda \geq 0$, there exist two measurable selection maps $\underline{T}_\lambda : \mathcal{S} \times \mathcal{A} \to \mathcal{A}$ and $\overline{T}_\lambda : \mathcal{S} \times \mathcal{A} \to \mathcal{A}$ such that for every $s \in \mathcal{S}, a \in \mathcal{A}$*

$$\underline{T}_\lambda(s,a) \in \arg\min_{a' \in \mathcal{D}_\lambda(s,a)} c(a,a'), \quad \overline{T}_\lambda(s,a) \in \arg\max_{a' \in \mathcal{D}_\lambda(s,a)} c(a,a'). \tag{5}$$

*2. If $\lambda^* > 0$, there exists $t^* \in [0,1]$ such that*

$$t^* \int_{\mathcal{S}} \int_{\mathcal{A}} c(a, \underline{T}_{\lambda^*}(s,a)) \mathrm{d}\pi(a|s) \mathrm{d}\rho_\pi(s) + (1-t^*) \int_{\mathcal{S}} \int_{\mathcal{A}} c(a, \overline{T}_{\lambda^*}(s,a)) \mathrm{d}\pi(a|s) \mathrm{d}\rho_\pi(s) = \varepsilon. \tag{6}$$

*3. There exists an optimal feasible policy $\tilde{\pi}$ for problem* (P) *defined for every $s \in \mathcal{S}$ by*

$$\tilde{\pi}(\cdot|s) := t^* \underline{T}_{\lambda^*}(s,\cdot)_\# \pi(\cdot|s) + (1-t^*) \overline{T}_{\lambda^*}(s,\cdot)_\# \pi(\cdot|s), \tag{7}$$

*where $t^*$ results from* (6) *if $\lambda^* > 0$ and $t^* = 0$ if $\lambda^* = 0$.*

Intuitively, Corollary 2 suggests that the optimal policy results from displacing the probability mass $\pi(a|s)$ to the maximizers of the $\lambda^*$-regularized advantage $\Phi_{\lambda^*}(s,a)$, where $\lambda^* \geq 0$ is the optimal dual solution. Since maximizers are generally not unique (i.e., $\mathcal{D}_{\lambda^*}(s,a)$ is not a singleton), one needs to balance between the *closest* (i.e., $\underline{T}_{\lambda^*}(s,a)$) and the *furthest apart* (i.e., $\overline{T}_{\lambda^*}(s,a)$) to satisfy the trust region constraint. In the special case $\lambda^* = 0$, the trust region constraint is either not active (i.e., the optimal policy lies within the trust region) or it does not affect the optimal trust region constraint (i.e., the optimal policy would lie at the boundary of the trust region even if the constraint is removed). In this case, it suffices to displace all probability mass to the closest maximizer of the advantage functions (i.e., $\underline{T}_0(s,a) \in \mathcal{D}_0(s,a)$). The complete proof of the results of this section is deferred to Appendix B.1.

*Remark* 1. Similar results were previously established in the literature in the context of DRO (e.g., see [8, Theorem 1] and [4, Theorem 1]). While these results closely inspire our proof, there is a major difference: in DRO, one seeks to evaluate the worst-case cost over an ambiguous set of probability distributions, expressed in terms of the optimal transport discrepancy. As such, the *average* optimal transport discrepancy in the trust region constraint is replaced by a single optimal transport discrepancy. Thus, one does not need to ensure the regularity of the problem with respect to the state (e.g., measurability of $\underline{T}_\lambda$ w.r.t. $s$). This is reflected in our assumption of *joint* continuity of the advantage in state and action and the state space being compact. To readily deploy existing results in DRO, one needs (i) to consider a single state only (i.e., $\mathcal{S} = \{s\}$) or (ii) to define a trust region *for each* state (at the price of infinitely many constraints). To transform (P) into a DRO, one might alternatively identify a policy as a probability measure over $\prod_{s \in \mathcal{S}} \mathcal{A}$, and hope to deploy standard duality arguments in DRO. However, the uncountable product of Polish spaces is not a Polish space, which makes all results in DRO, and more generally in optimal transport [37], inapplicable.

*Remark* 2. A similar result in the discrete case was presented in [33]. We highlight four major differences. First, in the discrete setting, problem (P) is a finite-dimensional linear optimization problem, for which strong duality holds. Thus, linear programming arguments can be used to derive the dual reformulation. In the continuous setting, the same proof strategy would require to mobilize the abstract machinery of infinite-dimensional linear programming [15]. Second, in the discrete setting, continuity and measurability of all functions are "for free". On the contrary, the continuous case imposes a careful analysis of these issues. Third, the optimal policy update [33] implicitly assumes that the set $\mathcal{D}_\lambda(s, a)$ (see (4)) is a singleton, which is rarely satisfied in practice. Fourth, as a byproduct of our proof, we show that (D) is a (one-dimensional) *convex* optimization problem, which can be solved efficiently via off-the-shelf solvers. This way, we do not need to resort to approximation techniques [33, Section 6.1] for the optimal dual multiplier.

**Discrete state-action spaces.** In the remainder of this section, we specialize our results to discrete (finite) state-action spaces (which trivially satisfy Assumptions 1 and 2). Without loss of generality, we represent the state and action spaces by $\mathcal{S} = \{s_1, \ldots, s_M\}$ and $\mathcal{A} = \{a_1, \ldots, a_N\}$ where $M$ and $N$ are two positive integers, and we describe any policy $\pi \in \Pi$ and its corresponding state-occupancy measure $\rho_\pi$ as discrete measures:

$$\pi(\cdot|s_i) = \sum_{j=1}^N \pi_{i,j} \delta_{a_j} \qquad \forall i \in \{1, \ldots, M\}, \qquad \rho_\pi = \sum_{i=1}^M \rho_i \delta_{s_i}, \qquad (8)$$

where $\rho_i, \pi_{i,j} \geq 0$ for every $i \in \{1, \ldots, M\}$, $j \in \{1, \ldots, N\}$, $\sum_{i=1}^M \rho_i = 1$ and $\sum_{j=1}^N \pi_{i,j} = 1$ for every $i \in \{1, \ldots, M\}$.[2] The analogous results to Theorem 1 and Corollary 2 are as follows.

**Corollary 3** (Dual formulation - discrete setting)**.** *Let $\varepsilon > 0$. For every policy $\pi \in \Pi$, the following strong duality result holds:*

$$\max_{\substack{t \in [0,1], \underline{b}_{i,j}, \overline{b}_{i,j} \in \mathcal{A}, \\ i \in \{1, \ldots M\}, j \in \{1, \ldots, N\}}} \left\{ \sum_{i=1}^M \rho_i \sum_{j=1}^N \pi_{i,j} \left( t A^\pi(s_i, \underline{b}_{i,j}) + (1-t) A^\pi(s_i, \overline{b}_{i,j}) \right) : \qquad \text{(discrete-P)} \right.$$

$$\left. \sum_{i=1}^M \rho_i \sum_{j=1}^N \pi_{i,j} \left( t c(a_j, \underline{b}_{i,j}) + (1-t) c(a_j, \overline{b}_{i,j}) \right) \leq \varepsilon \right\}$$

$$= \min_{\lambda \geq 0} \left\{ \lambda \varepsilon + \sum_{i=1}^M \rho_i \sum_{j=1}^N \pi_{i,j} \Phi_\lambda(s_i, a_j) \right\}. \qquad \text{(discrete-D)}$$

*In particular, let $\lambda^* \geq 0$ be a solution to* (discrete-D)*, and given for every $i \in \{1, \ldots, M\}$, $j \in \{1, \ldots, N\}$, select any*

$$\underline{b}_{i,j}^* \in \operatorname*{arg\,min}_{a' \in \mathcal{D}_{\lambda^*}(s_i, a_j)} c(a_j, a'), \qquad \overline{b}_{i,j}^* \in \operatorname*{arg\,max}_{a' \in \mathcal{D}_{\lambda^*}(s_i, a_j)} c(a_j, a'), \qquad (9)$$

*and let*

$$\underline{c} := \sum_{i=1}^M \rho_i \sum_{j=1}^N \pi_{i,j} c(a_j, \underline{b}_{i,j}^*), \qquad \overline{c} := \sum_{i=1}^M \rho_i \sum_{j=1}^N \pi_{i,j} c(a_j, \overline{b}_{i,j}^*).$$

---

[2]This representation is also valid beyond the finite state-action space setting when the policies and state-occupancy measures are empirical distributions with finitely many samples.

*Then, an optimal policy $\tilde{\pi}$ is given by*

$$\tilde{\pi}(\cdot|s_i) = \sum_{j=1}^{N} \pi_{i,j}\left(t^*\delta_{\underline{b}_{i,j}^*} + (1-t^*)\delta_{\overline{b}_{i,j}^*}\right), \qquad \forall i \in \{1,\dots,M\}, \tag{10}$$

*with $t^* = (\overline{c} - \varepsilon)/(\overline{c} - \underline{c}) \in [0,1]$ (and $t^* \in [0,1]$ if $\underline{c} = \overline{c} = \varepsilon$) if $\lambda^* > 0$ and $t^* = 0$ if $\lambda^* = 0$.*

The proof of this result stems from substituting the discrete measures as defined in (8) in problems (P) and (D), and observing that the images of the mappings $\underline{T}_{\lambda^*}$ and $\overline{T}_{\lambda^*}$ have finite support in the current setting. Notably, Corollary 3 directly provides an implementable algorithm for the policy update, circumventing the difficulty of the mixed-integer optimization problem (discrete-P): solving the one-dimensional convex program (discrete-D) provides the optimal Lagrange multiplier associated to the trust region constraint of the primal problem which can be directly used to compute the actions $\underline{b}_{i,j}^*$, $\overline{b}_{i,j}^*$ via (9), and thus the policy $\tilde{\pi}$ via (10).

*Remark* 3. The policy update suggested by (10) differs from the one in [33] (see [33, Theorem 1, (5)] where $f_s^*(i,j) \in \{0,1\}$ with their notation). Indeed, our policy update relies on "splitting the probability mass": the probability mass $\pi(\cdot|s_i)$ is displaced to $\underline{b}_{i,j}^*$ and $\overline{b}_{i,j}^*$ with weights $t^*$ and $1 - t^*$, respectively. This result is consistent with the Wasserstein DRO literature (e.g., see [4, Remark 2]). The result provided in [33] corresponds to the particular case where $\underline{b}_{i,j}^* = \overline{b}_{i,j}^*$ which amounts to supposing that the set $\mathcal{D}_\lambda(s_i, a_j)$ defined in (4) is a singleton. We provide further comments and examples in Appendix A.2 to illustrate the importance of this "mass splitting".

### 4.3 Policy improvement

In the next result, we show that our policy update leads to a monotonic improvement of the performance function $J$ up to the advantage function estimation error.

**Proposition 4** (Performance improvement)**.** *Let $\pi \in \Pi$. Consider solutions $\tilde{\pi}^* \in \Pi$ and $\lambda^* \geq 0$ of problems (P) and (D), respectively. If the true advantage function $A^\pi$ is approximated by some estimated advantage function $\hat{A}^\pi$ such that $\|A^\pi - \hat{A}^\pi\|_\infty < \infty$, then the following bound holds:*

$$J(\tilde{\pi}^*) \geq J(\pi) + \frac{\lambda^*}{1-\gamma}\int_{\mathcal{S}} C(\pi(\cdot|s), \tilde{\pi}^*(\cdot|s))\mathrm{d}\rho_{\tilde{\pi}^*}(s) - \frac{2\|A^\pi - \hat{A}^\pi\|_\infty}{1-\gamma}. \tag{11}$$

Proposition 4 indicates that optimal transport-based trust region policy optimization leads to monotonic improvement of the performance function when we have access to the true advantage function. The proof, postponed to Appendix B.2, of this result builds on the performance difference lemma (see (2)) and uses the closed-form expression of the optimal policy solving problem (P) as constructed in the proof of Theorem 1 (see Corollary 2). The analog of this result for a finite action space was proved in [33, Theorem 2, p. 5]. To the best of our knowledge, this result is novel for the continuous state-action space setting.

## 5    Practical Optimal Transport Trust Region Policy Optimization Algorithm

In this section, we use the duality results on Section 4 to derive a practical algorithm for OT-TRPO. Herein, we restrict the policy search set $\Pi$ to the set of policies $\pi_\theta$ parametrized by a vector $\theta \in \mathbb{R}^d$ for some integer $d > 0$. We require the policy parametrization to be continuously differentiable with respect to $\theta$ (for every state and action). This way, we simultaneously cover the direct parametrization (for which Corollary 3 directly provides a policy update) as well as commonly used policy parametrizations (e.g., softmax and the Gaussian policies). Accordingly, the dual problem (D) can be reformulated as follows for every $\theta \in \mathbb{R}^d$:

$$\min_{\lambda \geq 0} G(\lambda, \theta) := \lambda\varepsilon + \int_{\mathcal{S}}\int_{\mathcal{A}} \max_{a' \in \mathcal{A}}\{A^{\pi_\theta}(s, a') - \lambda c(a, a')\}\mathrm{d}\pi_\theta(a|s)\mathrm{d}\rho_{\pi_\theta}(s). \tag{12}$$

Given a current policy represented by the vector $\theta$, we first solve the one-dimensional convex problem (12) to obtain its solution $\lambda^*$. Then, we use the optimal dual multiplier $\lambda^*$ to derive the optimal policy update within the trust region. The procedure is summarized in Algorithm 1. Depending on the parametrization of the policy, the steps of Section 5 are as follows:

---
**Algorithm 1** OT-TRPO.
---
1: Initialize $\pi_{\theta_0}$
2: **for all** $t = 0, 1, \ldots$ **do**
3:     Estimate $A^{\pi_{\theta_t}}$ and $\rho_{\pi_{\theta_t}}$.
4:     Compute $\lambda^* \in \operatorname{argmin}_{\lambda \geq 0} G(\lambda, \theta_t)$.
5:     Update $\theta_t$ to $\theta_{t+1}$ using $\lambda^*$.
6: **end for**
---

**Algorithm 1 - step 3.** In the discrete setting, the visitation frequency is estimated via Monte Carlo methods. In the continuous case, we weight every visited state equally. We propose three ways to estimate the unknown advantage function via samples[3]:

1. Monte Carlo methods or TD-learning (for discrete settings only);
2. General Advantage Estimation (GAE) [30], using a neural network to approximate the value function like in standard actor-critic methods; and
3. Direct estimation via non-linear approximators (e.g., using directly a neural network for the advantage function).

**Algorithm 1 - step 4 (evaluation of $G$).** Depending on the setting, we propose various ways to evaluate $G(\lambda, \theta)$. They all apply to both continuous and discrete states.

1. *Finite actions*: Since the maximization in (12) is over finitely many actions, we can directly evaluate (12) for any $\lambda \geq 0$.
2. *Gaussian policy parametrization*: With $m(s)$ being the mean of the Gaussian policy (with fixed variance), we approximate $G(\lambda, \theta)$ by

$$G(\lambda, \theta) \approx \begin{cases} \lambda \varepsilon + \sum_{s \in \hat{\mathcal{S}}} \max_{a' \in \{a, m(s)\}} \{A^{\pi_\theta}(s, a') - \lambda c(m_\theta(s), a')\} \rho_{\pi_\theta}(s) & \text{if } A^{\pi_\theta} \text{ via GAE,} \\ \lambda \varepsilon + \sum_{s \in \hat{\mathcal{S}}} \max_{a' \in \hat{\mathcal{A}}(s)} \{A^{\pi_\theta}(s, a') - \lambda c(m_\theta(s), a')\} \rho_{\pi_\theta}(s) & \text{if } A^{\pi_\theta} \text{ via NN,} \end{cases}$$

   where $\hat{\mathcal{S}}$ are the states visited in the trajectory and $\hat{\mathcal{A}}(s)$ is a (possibly state-dependent) collection of samples from $\mathcal{A}$.
3. *Arbitrary policy parametrization*: For a neural network approximation of the advantage function, we approximate $G(\lambda, \theta)$ by

$$G(\lambda, \theta) \approx \lambda \varepsilon + \sum_{s \in \hat{\mathcal{S}}} \sum_{a \in \hat{\mathcal{A}}_1(s)} \max_{a' \in \hat{\mathcal{A}}_2(s)} \{A^{\pi_\theta}(s, a') - \lambda c(a, a')\} \pi_\theta(a|s) \rho_{\pi_\theta}(s),$$

   where $\hat{\mathcal{S}}$ are the states visited in the trajectory and $\hat{\mathcal{A}}_i(s)$ are (possibly state-dependent) collections of samples from $\mathcal{A}$.

**Algorithm 1 - step 4 (solving for $\lambda^*$).** Since (D) is a one-dimensional convex optimization problem, $\lambda^*$ can be found using any solver for convex optimization problems.

**Algorithm 1 - step 5.** Update the parameter vector $\theta$.

1. *Direct parametrization (finite spaces)*: Update $\theta$ according to (10) and (9).
2. *Direct parametrization via policy network (continuous states, discrete actions)*: Use (10) and (9) to compute the optimal policy update at the visited states, denoted by $\pi_{\theta_t}^*$. Then, update the policy network by performing gradient descent on the loss $L(\theta) = \sum_{s \in \hat{\mathcal{S}}} \rho_{\pi_{\theta_t}}(s) \big\| \pi_\theta(\cdot|s) - \pi_{\theta_t}^*(\cdot|s) \big\|^2$ to steer $\pi_\theta$ towards the optimal policy update $\pi_{\theta_t}^*$ *within* the trust region.
3. *Arbitrary policy parametrization*: Since there are infinitely many actions, the computation of the maximization is computationally demanding, and so Corollary 2 cannot be directly utilized for the policy update. Yet, we can update the policy via gradient ascent. The intuition is as follows: according to Corollary 2, the optimal policy update attains the maximum $\max_{a' \in \mathcal{A}} \{A^{\pi_{\theta_t}}(s, a') - \lambda^* c(a, a')\}$ at each state. Thus, we can steer the policy $\pi_\theta$ to maximize

$$\theta \mapsto \sum_{s \in \hat{\mathcal{S}}} \int_{\mathcal{A}} \max_{a' \in \mathcal{A}} \{A^{\pi_{\theta_t}}(s, a') - \lambda^* c(a_\theta, a')\} \mathrm{d}\pi_\theta(a_\theta|s) \rho_{\pi_{\theta_t}}(s).$$

   In the particular case of a Gaussian policy with parametrized mean (and fixed variance), combined with GAE estimate of the advantage function, one can maximize

$$\theta \mapsto \sum_{s \in \hat{\mathcal{S}}} \max \{A^{\pi_{\theta_t}}(s, a') - \lambda^* c(m_\theta(s), a'), 0\} \rho_{\pi_{\theta_t}}(s).$$

---

[3]In the experiments reported in the main paper we used the first and the second method for discrete and continuous environments, respectively. In Appendix A.5 we further comment on the different methods.

Intuitively, this update implicitly estimates the transport maps $\overline{T}_{\lambda^*}$ and $\underline{T}_{\lambda^*}$, which are needed for the optimal policy update. This way, we steer the policy network towards the optimal policy update *within* the trust region. Among others, this policy update allows for the following interpretation: imposing an optimal transport-based trust region constraints is, at least formally, equivalent to maximizing a *regularized* advantage function, where the value of the regularization $\lambda^* \geq 0$ is based on the transport cost $c$ and the radius of the trust region $\varepsilon$.

## 6 Experiments and Insights

In this section, we evaluate the performance of OT-TRPO across a variety of environments [5, 36] of increasing complexity, ranging from discrete to continuous settings. We compare it to the classical TRPO [29, 9] and PPO [31, 9], with Advantage Actor Critic (A2C) [17, 9], with the recent approaches leveraging the Wasserstein distance, BGPG [22] and WNPG [20] (in continuous settings), and with WPO [33] (in discrete settings). The training curves are shown in Fig. 1; see Appendix A.3 for implementation details, Appendix A.6 for further details on the experimental results, and Appendix A.4 for an ablation study on our algorithm.

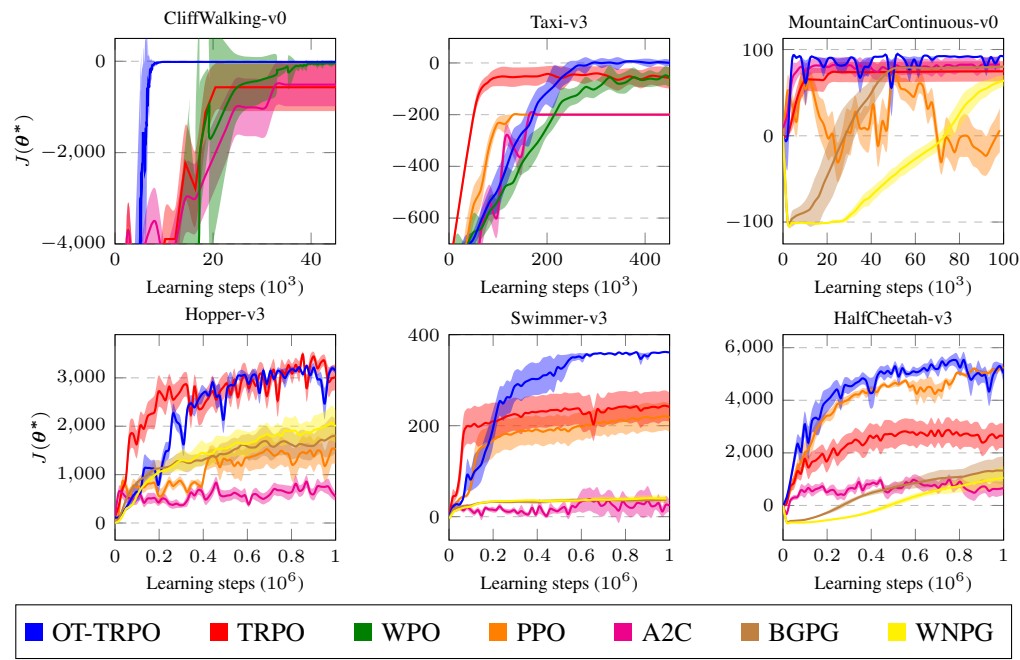

Figure 1: Cumulative rewards during the training process in different environments. The shaded area represents the mean $\pm$ the standard deviation across 10 independent runs. Every policy evaluation in each run is averaged over 10 sampled trajectories.

Our approach is shown to consistently improve over the other algorithms: OT-TRPO leads to larger final returns, with lower variance, and only in few cases at the expense of a slightly slower learning curve. Four remarks are in order. First, the performance gain of OT-TRPO compared to BGPG and WNPG confirms that trust regions help stabilize training, as already observed in [29]. Second, optimal transport discrepancies induce a more natural notion of "closeness" between policies compared to the KL divergence (e.g., see [3, Example 2.1]). For instance, in CliffWalking-v0, consider the optimal policy $\pi^*$ and the candidate policy $\pi$ depicted below, which differ at one state only (see figure). The optimal transport discrepancy between $\pi$ and $\pi^*$ is $\rho_\pi(s)C(\pi(\cdot|s), \pi^*(\cdot|s)) = \rho_\pi(s)c(\text{Down}, \text{Right})$. When using the KL divergence, instead, the discrepancy is infinite, since the two policies do not share the same support. In particular, if initialized with $\pi$, TRPO cannot converge to the optimal policy, regardless of the radius of the trust region. Third, OT-TRPO improves on WPO, in two ways: (i) it leads to superior performances of the trained policies and (ii) it does not violate

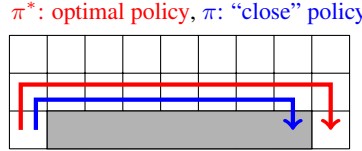

$\pi^*$: optimal policy, $\pi$: "close" policy

the trust region constraint (which, e.g., in Taxi-v3 is the case for 72% of the updates of WPO). This performance improvement results from the "mass splitting" (see Remark 3), which is the only difference between the two algorithms (in discrete settings). Fourth, in continuous settings, the performance of OT-TRPO is aligned with the best-performing alternative approach. In the environment Swimmer-v3, it even yields an improvement of more than 50% in the performance of the trained agent.

## 7    Conclusion and Future Work

We studied trust region policy optimization for continuous state-action spaces whereby the trust region is defined in terms of a general optimal transport discrepancy. Our analysis bases on a one-dimensional convex dual reformulation of the optimization problem for the policy update which (i) enjoys strong duality and (ii) directly characterizes the optimal policy update, bypassing the computational burden of evaluating optimal transport discrepancies. Moreover, we show that the policy update can yield a monotonic improvement of the performance index. Empowered by our theoretic results, we propose a novel algorithm, OT-TRPO, for trust region policy optimization with optimal transport discrepancies. We evaluate its performance across several environments. Our results reveal that trust regions defined by optimal transport discrepancies can offer advantages over the KL divergence or non-trust region methods.

There are several research directions that merit further investigation. We highlight two. First, transport costs provide us actionable knobs to shape the geometry of the trust region, and can be used to encode prior knowledge on the environment or preferred exploration strategies. Second, we would like to study the convergence properties of the proposed algorithm.

## Acknowledgements

This project has received funding from Google Brain, Swiss National Science Foundation under the NCCR Automation (grant agreement 51NF40_180545), and it was partially supported by the ETH AI Center.

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
