# OpenReview forum: "Trust Region Policy Optimization with Optimal Transport Discrepancies: Duality and Algorithm for Continuous Actions"
_NeurIPS.cc/2022/Conference — NeurIPS 2022 Accept_

### Official Review · Reviewer_JmUx · 2022-07-08

**Rating:** 7
**Confidence:** 4
**Soundness:** 3 good
**Presentation:** 3 good
**Contribution:** 4 excellent

**Summary:**

The paper proposes to use optimal transport discrepancies as an alternative to the Kullback-Leibler divergence for Trust Region Policy Optimisation (TRPO). The authors provide the intractable optimisation problem (P), and then propose a dual formulation (D) of the optimal transport problem which is tractable. They base the proposed algorithm upon this dual formulation and perform the analysis of the method.


**Questions:**

Questions:

**Question 1:** While the analysis in the paper gives substantial detail on experimental comparison against other methods, as well as on a choice of optimal transport distance, it is still important to give more details on the failure modes. The closest I could get to that were the results from Hopper (Figure 1), where the model performs comparably with standard TRPO. Would be interesting to see if there are any other examples when TRPO works pretty much as well or outperforms the model, and what could be the reason behind it? While seeing that the experimental results show convincing experimental advantage, it looks like covering more limitations could be helpful in understanding the potential downsides of the method.

**Question 2:**
In line 664 onwards, authors write:
 “In continuous action spaces, we also experimented with training a neural network that approximates  the advantage function of the current policy using Generalized Advantage Estimators. Our experimental results indicate that policy optimization with such a neural network approximate does not perform comparably to the single sample estimation of the objective. In general, we observed that (1) simultaneous training of an advantage network and the OT-TRPO policy update could be unstable, and (2) in the case of convergence, OT-TRPO with neural advantage function approximation converges  to a suboptimal policy. We hypothesize that the OT-TRPO objective is sensitive to biased value estimates due to the infimum over the action space in the neighborhood of the mean of the policy.  The observation that the optimal GAE hyperparameter in the single-sample advantage estimation  was very close to 1 (i.e., an unbiased Monte Carlo estimate) in our experiments supports this idea. “Would be really good to explicitly state, for the experiments in the main text and in the appendix, which policy parameterisations and which advantage function estimates correspond to which experiment. To support the claim, is it possible, for example, to plot the comparison graphs for different types of estimates?

**Question 3: **
In section A.5, I haven’t found enough information on the details of implementation of the Monte-Carlo estimate, could the authors clarify on this to ensure reproducibility?

**Limitations:**

Societal impact limitations: I think the authors sufficiently addressed them, especially given that the problem statement of reinforcement learning in simulated environments, the datasets and similar methods are well-known as detailed in Section 2 of the paper.

Technical limitations: as outlined in questions, more experimental evidence on the failure modes could be given (see Question 1 for the details).

**Strengths And Weaknesses:**


Strengths:
- The problem statement is interesting and gives insight into optimal transport formulation of the TRPO. The mathematical implementation is solid, and the idea of dual problem formulation to circumvent the intractability is theoretically well grounded.
- The work is well placed within the current literature, and the authors contrast it with the existing trust regions and optimal transport-based methods such as Song et al (2022).

Weaknesses:
- While the paper is sound, in the experimental analysis, it could be good to show the failure modes (see below) (see question 1 for the details below)
- It would be also good to show the impact of different policy parameterisations as described in Algorithm 1, Step 5, as well as different estimates of the advantage function (Step 5); it is also important to more clearly outline which experiment uses which policy parameterisation and the advantage function (see question 2)

---

> ### Author Response · Authors · 2022-08-02
> **Answer to Reviewer JmUx**
>
> We thank the reviewer for the positive feedback and constructive suggestions. We address the raised questions separately below. We hope that our answers clarify the doubts and address the concern of the reviewer, increasing the strength of our contribution.
>
> 1. We thank the reviewer for the important question: We agree that understanding the limitations is crucial. For this reason, we appositely added some comments in Appendix A.5 and A.6. We hope that it will address the reviewers' doubts and strengthen and clarify our contribution.
> From the provided benchmarks, the algorithm seems to suffer from the following limitations:
> -  In some environments, like Taxi, Hopper, and Swimmer, TRPO seems to learn much faster than our algorithm. This is closely related to the choice of trust-region. With OT-TRPO, moving the “probability mass” is done at the cost $c(a_1, a_2)$. In TRPO, there is no notion of transport cost in the action space: all the actions are at the same “distance”. I.e., they differ only based on the probability of using them when at a state $s \in \mathcal{S}$; namely $\log{}(\pi(a_1|s) / \pi(a_2|s))$. Whenever $c(a_1, a_2) > \log{}(\pi(a_1|s) / \pi(a_2|s))$, the convergence is faster with TRPO compared to OT-TRPO. Clearly, one could design a transport cost that speeds up the convergence (a “smaller” one), at the price of a potentially less stable and robust behavior of the algorithm. The study of the “optimal” choice of transport cost is indeed an interesting question for future research. We discuss this further in Appendix A.6.
> - Closely related is our additions regarding the reviewers second question. We have added ablation studies regarding advantage estimation with neural networks, which seems to perform worse compared to a low-biased GAE estimate. As we mention in our paper, this could be due to the sensitivity of the dual infimum problem to biased estimates. We leave it as a future research question if better parameterizations are possible for functional advantage approximation.
> Albeit in the various experimented environments OT-TRPO performs comparably or better than TRPO, we can construct a corner case where the proposed loss function for continuous action spaces does not converge to the optimum for any arbitrary initialization. On the contrary, TRPO would not encounter such a problem. We discuss it further in Appendix A.5.
>
> 2. We have amended the text to clarify that the results presented in the paper use a generalized advantage estimator and not an advantage value estimated by a neural network. We thank the reviewer for suggesting this weakness, which we hope we have addressed. Similarly, we have now added to section A.5 our experimental results with respect to different advantage estimation methods, backing up our claims. We also added an ablation study demonstrating the dependence on the $\lambda_{\mathrm{GAE}}$ hyperparameter (see [Schulman, 2016]).
>
> 3. We thank the reviewer for pointing out the unclear explanation. We added more details in the revised version, with the hope to enhance reproducibility. We remark that we also provide the code for all the algorithms proposed in the supplementary material.
> - In discrete spaces, the visitation frequency is computed counting the number of visits to a specific state, and normalizing. The advantage function is estimated in the following way:
> i. We collect a set of trajectories $\mathcal{T}$, and we initialize $Q(s, a) = 0$;
> ii. For all trajectory $\tau \in \mathcal{T}$, and every (s_t, a_t, r_t, s_{tt}, a_{tt}) in $\tau$, we update $Q$ as
> $Q(s_t, a_t) = (1 - \alpha) Q(s_t, a_t) + \alpha (r_t + \gamma Q(s_tt, a_tt))$.
> - In continuous spaces, the visitation frequency is the uniform distribution over the states visited (as no state is visited twice almost surely in continuous spaces). The advantage function is instead estimated using the GAE, with the implementation provided by stable baselines (https://stable-baselines.readthedocs.io/en/master/ ). The advantage estimation in continuous spaces (with comparisons between different methods) is discussed thoroughly in the paragraph "Sensitivity to the advantage estimation”, Appendix A.4.

---

> > ### Comment · Reviewer_JmUx · 2022-08-05
> > **Thank you for the responses**
> >
> > Thank you for addressing the questions, as well as other other reviewers' questions (for example, the answer on hVuN's Question 1 addresses similar concerns on limitations). The new updates help better understand the limitations and clarify upon the Monte-Carlo estimate (Q3). The score (7: Accept) remains the same.

---

### Official Review · Reviewer_71r1 · 2022-07-11

**Rating:** 6
**Confidence:** 3
**Soundness:** 3 good
**Presentation:** 4 excellent
**Contribution:** 3 good

**Summary:**

The paper under review studied the problem of trust region policy optimization(TRPO).
TRPO aims to optimize the expected return through stable policy update, where the
stability is control by only allowing update within certain region around the current policy
in each iteration. Hence one needs to pick a norm to measure distances between policies.
In this paper, the authors proposed to use OT distance.
Allowing the continuous setting for action and state spaces,
they derive the dual of the proposed OT-TRPO and proved strong duality,
base on which, an update algorithm was proposed.
Moreover, by establishing eq (11), the authors demonstrated the monotonic improvement of the proposed method.
In addition, the authors illustrated the efficiency of OT-TRPO through simulations.



**Questions:**

Q1: In section 4.3, the authors stated that eq(11) shows monotonicity of the update.
However, to guarantee $J(\tilde{\pi}^*) > J(\pi)$, one needs to show that the sum of the last two terms in eq(11) is positive.
It is not clear why the sum is always positive. Would the authors explain a bit further on it?

Q2: Regarding the experiments: what are the choices of costs and other parameters? Are the results sensitive to these choices?
What is the running time comparing to other methods? In particular, step 4 and step 5 in Algo1.

Q3: Is it possible to empirically validate that the proposed algorithm actually approximates the solution of problem (P)?
For example, check for a small discrete setting where OT solution can be computed?

Q4: Does duality proved in Thm 1 holds for Sinkhorn divergence (instead of classical OT without the entropy term)?


**Limitations:**

The authors mentioned the convergence properties of the proposed algorithm is left open.
I agree with the authors that proving converges and calculation of convergence rate
is an essential step that is currently missing.

**Strengths And Weaknesses:**

The paper is well written and easy to follow.
The idea of using optimal transport discrepancy is natural and interesting.
The proposed update algorithm is nicely stated and and properly motivated.
The fact that it bypass the actual computation of the forward OT has many potential applications.
The theoretical results are sounds besides some questions listed below.

---

> ### Author Response · Authors · 2022-08-02
> **Answer to Reviewer 71r1**
>
> We thank the reviewer for the positive feedback and constructive suggestions. We answer the raised questions point by point below and in the reviewed version, with the hope that this will increase the strength of our contribution.
>
> 1. Whether or not the sum is positive depends on the estimated advantage function. If the advantage function is estimated with sufficient precision (i.e., with sufficiently many samples), then the second term is small (as the approximation error goes to zero with increasing number of samples) and the sum is indeed positive (as the optimal transport discrepancy is). This is in line with existing work: e.g., Theorem 1, Schulman et al. 2015, Theorem 2, Song et al. 2022.
>
> 2. We believe this is a legitimate and important question, which we aimed at answering carefully in Appendix A.
> - The choices of costs and parameters are discussed in Appendix A.3.1 (Implementation details). In particular, in the paragraph “Discrete settings”, we mention that the selected transport cost is a binary distance.The parameters for such settings are summarized in Table 1. The “Continuous settings” parameters are gathered in Table 2; the transportation cost is the squared euclidean distance.
> - The sensitivity to these choices (trust region radius, the transportation cost and the advantage function estimate) is discussed in the ablation study in Appendix A.4.
> - The training time comparisons are summarized in Table 5. We take the chance to highlight what we believe to be one strength of our algorithm: The line search of TRPO with optimal transport costs constraints would be computationally prohibitive. Similarly, the work exploiting optimal transport constraints (e.g., BGPG, WNPG) is extremely slow compared to other algorithms (e.g., PPO, TRPO, A2C). Instead, the proposed method is as computationally efficient as the latter class of algorithms.
> - We extended the discussion in Appendix A.3.1. For space reasons, we refer the reviewer to the updated section.
>
> 3. We complemented the appendix to answer thoroughly to the interesting question; please refer to A.5 for a complete answer.
> In discrete settings, the update is theoretically guaranteed to be exact, and the numerical approximations introduced by the solver of the dual problem are the only source of approximations.
> For continuous state spaces, a few remarks are in order about Step 5:
> - The loss function in Option 2 drives the policy update towards a set of parameters that behaves like the optimal policy. With the information available and for the policy parametrization chosen, this is the best one can do. It is however limited to discrete action spaces.
> - The loss function in Option 3 allows us to circumvent the computation of the optimal transport cost constraint and deploy the algorithm in practical instances. More importantly, such loss is theoretically baked by a strong duality result. We added a discussion on this in Appendix A.5, with a numerical/analytical example, as suggested by the reviewer. The intuition behind this loss function is to increase the probability mass where the regularized advantage function has its maxima. However, in general, the solutions are multiple, while the gradient descent will possibly converge to one: One limitation of this cost function is related with the mass-splitting issue. Empirically, the proposed loss function performs well, and allows the deployment of optimal transport trust-region methods in continuous spaces. Further work will focus on understanding how to implicitly describe transport \emph{plans}, rather than transport maps. Moreover, the example shows that depending on the parametrization, it is possible to experience zero gradient when the optimal policy is within the trust region. This suggests an iterative decrease scheme for the trust region radius.
>
> 4. Our duality results do not directly apply to Sinkhorn divergence (or entropy-regularized optimal transport discrepancies in general). For instance, our proof of weak duality (Proposition 5 in Appendix B1) leverages the Kantorovich duality, which is optimal transport specific. However, duality results for entropy-regularized optimal transport [Terjèk and González-Sánchez 2022], and recent results in DRO [Gao et al. 2017] suggest that our proof can be adapted to such constraints. Having said this, we remark that Sinkhorn divergence was successful in mitigating the burden of optimal transport computations. The proposed algorithm does not rely on any of such, and it is thus unclear whether regularized optimal transport discrepancies could yield any benefit in this context.
>
> We thank again the reviewer to point out the major contributions of our work: (1) we proved strong duality of the optimal transport trust region policy optimization problem, (2) we show monotonic improvement of the performance function, (3) we propose a novel PO algorithm for continuous spaces. Proving convergence rate results will definitely be addressed in a future work.

---

> > ### Comment · Reviewer_71r1 · 2022-08-09
> > **Thank you for the response**
> >
> > Thank you for addressing my questions, and others too.
> > It would be nice if the authors could add some pointers in the main text to the appropriate sections in the appendix.

---

> > > ### Author Response · Authors · 2022-08-09
> > > **Thank you for the feedback**
> > >
> > > We thank the reviewer for the feedback. We will include pointers to the sections of the appendix in the final version of the paper.

---

### Official Review · Reviewer_hVuN · 2022-07-12

**Rating:** 5
**Confidence:** 3
**Soundness:** 3 good
**Presentation:** 3 good
**Contribution:** 3 good

**Summary:**

This paper studies the trust region policy optimization (TRPO) problem, by replacing the KL divergence policy discrepancy constraint in TRPO with optimal transport discrepancies, such as Wasserstein distance. Considering the optimal transport discrepancy region, a dual form of the constrained problem has been developed, which contains a Lagrange multiplier \lambda. Find the optimal \lambda^* at each step only needs to solve a one-dimensional convex optimization problem, which can be solved using off-the-shelf solvers, instead of line-search in TRPO. Then, the optimal \pi can be computed using \lambda^* with closed-form solution. Experiments on both discrete and continuous settings are conducted.

**Questions:**

1. Are there any explanations that for many environments, such as Taxi, Hopper, and Swimmer, TRPO learns much fasters at early stages compared to OT-TRPO? Does these environments share some similarity in dealing with the trust regions?

2. WPO can be extended to deal with continuous actions as studied in the original paper, while it is ignored here. For example, in Hopper, WPO can reach much higher scores if we count for the same amount of learning steps. In Figure 1, it seems TRPO can achieve very competitive performance, while this differs from the results reported in BGPG and WNPG. Would the implementation of TRPO has some particular treatment here?

3. Would an efficiency comparison with the line-search procedure in TRPO be possible? Since one important benefit using the optimal transport discrepancy is that the optimal policy at each step has a closed-form solution to avoid the computational line-search.

4. Line 341: “it leads to superior asymptotic performance”. I do not think the sentence is rigorous, since when you refer to asymptotic performance you consider nearly infinite samples while the learning steps in the experiments are of very small scale considering current RL problems.

5. Minor typos:
Line 338: “not do” -> “do not”


**Limitations:**

Some experimental results of the compared methods are not clear enough. Benchmark environments should be enriched to thoroughly evaluate the empirical performance of OT-TRPO.

**Strengths And Weaknesses:**

Pros:

The paper is clearly written. It proposes a new dual form of trust region PO problem using optimal transport discrepancy, where the solution can be computed efficiently compared to line-search using KL divergence.


Cons:

The compared methods have included many closely related works in trust region PO, however, the benchmark environments are somehow limited and insufficient. Since the OT-TRPO is derived for both discrete and continuous actions, to give a thorough support for the effectiveness of OT-TRPO, it is recommended to evaluate on some standard RL benchmarks such as the Atari suite, and Mujoco families. The considered tasks, such as CliffWalking, Taxi, MountainCar, Swimmer, etc., are relatively simple environments, in which the learning steps are relatively small to obtain good performances for general algorithms.

---

> ### Author Response · Authors · 2022-08-02
> **Answer to Reviewer hVuN**
>
> We thank the reviewer for the feedback and constructive comments. We answer the raised questions below. We hope that our answers clarify the doubts and address the concern of the reviewer, increasing the strength of our contribution.
>
> 1. We thank the reviewer for the interesting question. We added some comments in Appendix A.6, with the hope this will strengthen and clarify our contribution.
> In trust-region based methods, the trust-region definition provides a trade-off between speed and stability of the convergence. An interpretation of the observed phenomenon follows:
> With OT-TRPO, moving the “probability mass” is done at the cost $c(a_1, a_2)$. In TRPO, there is no notion of transport cost in the action space: all the actions are at the same “distance”. I.e., they differ only based on the probability of using them when at a state $s \in \mathcal{S}$; namely $\log{}(\pi(a_1|s) / \pi(a_2|s))$. Whenever $c(a_1, a_2) > \log{}(\pi(a_1|s) / \pi(a_2|s))$, the convergence is faster with TRPO compared to OT-TRPO. Clearly, one could design a transport cost that speeds up the convergence (a “smaller” one), at the price of a potentially less stable and robust behavior of the algorithm.
> As an example, in the Taxi environment there are $6$ actions. The proposed method with a binary distance incurs a cost of $1$ to bring a state from a uniform probability distribution over the actions to a deterministic one (which for most states is the case in the optimal policy). Instead, TRPO considers such policies only $\log{}(6) < 1$ away.
> Another reason can be found in the lack of symmetry of the KL divergence. We further discuss this in Appendix A.6 for space reasons.
>
> 2. The algorithm (WPO) presented in Song et al. 2022 works in continuous action spaces via discretization (Section 6.4). Such an approach scales poorly with the dimension of the action space, and was thus excluded from the comparison. Furthermore, the results regarding that section are not reproducible for the lack of parameters and implementation. The code can be found at  https://github.com/efficientwpo/EfficientWPO, but it does not include the discretization part and the parameters are not provided in the paper. To the best of our effort, our implementation of WPO in continuous actions is extremely slow, due to the high dimensionality of the action space. We also remark that the version of Hopper they refer to is Hopper-v2, whereas our experiments are conducted in the newest and more difficult Hopper-v3.
> Regarding Figure 1, we were also surprised by the performance difference. In Song et al. 2022, they use their own implementation of TRPO, whereas we use the state-of-the-art implementation, provided by stable baselines (see https://stable-baselines.readthedocs.io). We also remark that BGPG and WNPG were evaluated on older (easier) environments.
>
> 3. The implementation of both OT-TRPO and TRPO are based on https://stable-baselines.readthedocs.io. They differ indeed only in the policy update step. In Table 5, we collected the training time of the various algorithms. One could then compare the timings of TRPO and OT-TRPO to compare the line-search of the former with the exact update of the latter. From such analysis it would appear that the method does not yield computational benefits. However, the line search of TRPO is performed on a KL divergence constraint, which is much faster to compute than an optimal transport cost constraint. With this regard, a fairer comparison would be between the exact policy update and a line search on an optimal transport cost constraint. This comparison would then strongly highlight the benefits of the method proposed, as such line search is not computationally feasible, whereas our algorithm training time is in line with e.g. PPO, TRPO.
>
> 4,5. Thank you for pointing it out. We changed it in the new version.
>
> We thank the reviewer for suggesting how to strengthen our paper. We acknowledge that additional benchmarks would better assess our contribution. However, Atari environments are substantially different from the experimented ones, as they require a convolutional neural network. We will work to have some Atari experiments for the camera ready version, but we are unable to provide them in this review due to the time limit. However, we would like to note that the environments in the original submission include multiple Mujoco control problems used as challenging benchmarks (such as Swimmer, Walker, and HalfCheetah) in previous work. Finally, aiming at providing a better understanding of the strength and weaknesses of the algorithm (and following the other reviewers suggestions), we included additional analysis of the policy update in A.5 and additional experiments on the advantage estimation in A.4. We hope that the reviewer will find these additions insightful.

---

### Meta-Review · Area_Chair_Tk2n · 2022-09-09

**Recommendation:** Accept
**Confidence:** Certain

**Metareview:**

The paper studies trust region optimization but replace the typical KL divergence with optimal transport distance - which is a natural and meaningful generalization. The authors provided a tractable algorithm by using optimization duality, and provide experimental results on control tasks. All reviewers appreciate the main idea is novel and interesting, and further the paper is very written and easy to understand. Some reviewer have questions about the theory and experiments. The authors largely address the reviewers' comments during rebuttal, and all authors are in favor of acceptance.

**Award:**

No

---

### Decision · Program_Chairs · 2022-09-14

Accept